# HINT-BASED TRAINING
# FOR NON-AUTOREGRESSIVE TRANSLATION

## ABSTRACT

Machine translation is an important real-world application, and neural network-based AutoRegressive Translation (ART) models have achieved very promising accuracy. Due to the unparallelizable nature of the autoregressive factorization, ART models have to generate tokens one by one during decoding and thus suffer from high inference latency. Recently, Non-AutoRegressive Translation (NART) models were proposed to reduce the inference time. However, they could only achieve inferior accuracy compared with ART models. To improve the accuracy of NART models, in this paper, we propose to leverage the *hints* from a well-trained ART model to train the NART model. We define two *hints* for the machine translation task: hints from hidden states and hints from word alignments, and use such hints to regularize the optimization of NART models. Experimental results show that the NART model trained with hints could achieve significantly better translation performance than previous NART models on several tasks. In particular, for the WMT14 En-De and De-En task, we obtain BLEU scores of 25.20 and 29.52 respectively, which largely outperforms the previous non-autoregressive baselines. It is even comparable to a strong LSTM-based ART model (24.60 on WMT14 En-De), but one order of magnitude faster in inference.

## 1 INTRODUCTION

Neural machine translation has attracted much attention from the research community (Bahdanau et al., 2014; 2016; Kalchbrenner et al., 2016; Gehring et al., 2016) and has been gradually adopted by industry in the past several years (Wu et al., 2016). Despite the huge variety of model architectures (Bahdanau et al., 2014; Gehring et al., 2017; Vaswani et al., 2017), given a source sentence $x = (x_1, ..., x_{T_x})$ and a target sentence $y = (y_1, ..., y_{T_y})$, most neural machine translation models decompose and estimate the conditional probability $P(y|x)$ in an universal autoregressive manner:

$$P(y|x) = \Pi_{t=1}^{T_y} P(y_t|y_{<t}, x), \tag{1}$$

where $y_{<t}$ represents the first $t-1$ words of $y$. During inference, given an input sentence, those models generate the translation results sequentially, token by token from left to right. We call all such models *AutoRegressive* neural machine Translation (ART) models. A state-of-the-art ART model, Transformer (Vaswani et al., 2017), is shown in the left part of Figure 1.

A well-known limitation of the ART models is that the inference process can hardly be parallelized, and the inference time is linear with respect to the length of the target sequence. As a result, the ART models suffer from long inference time (Wu et al., 2016), which is sometimes unaffordable for industrial applications. Consequently, people start to develop *Non-AutoRegressive* neural machine Translation (NART) models to speed up the inference process (Gu et al., 2017; Kaiser et al., 2018; Lee et al., 2018). These models use the general encoder-decoder framework: the encoder takes a source sentence $x$ as input and generates a set of contextual embeddings and predicted length $T_y$; conditioned on the contextual embeddings, the decoder takes a transformed copy of $x$ as input and predicts the target tokens at all the positions independently in parallel according to the following decomposition:

$$P(y|x, T_y) = \Pi_{t=1}^{T_y} P(y_t|T_y, x). \tag{2}$$

While the NART models achieve significant speedup during inference (Gu et al., 2017), their accuracy is considerably lower than their ART counterpart. Most of the previous works attribute the

poor performance to this unavoidable *conditional independence* assumption of the NART model. To tackle this issue, they try to improve the expressiveness and accuracy of the decoder input in different ways: Gu et al. (2017) introduce *fertilities* from statistical machine translation models into the NART models, Lee et al. (2018) base the decoding process of their proposed model on an iterative refinement process, and Kaiser et al. (2018) take a step further to embed an autoregressive submodule that consists of discrete latent variables into their model. Although such methods provide better expressiveness of decoder inputs and improve the final translation accuracy, the inference speed of these models will be hurt due to the overhead of the introduced modules, which contradicts with the original purpose of introducing the NART models, i.e., to parallelize and speed up neural machine translation models.

Different from previous works that develop new submodules for decoder input, we improve the translation model from another perspective. We aim to provide more guided signals during optimization. That is, we do not introduce any new prediction submodule but introduce better regularization. The reason we tackle the problem from this perspective lies in two points: First, the encoder input (source words) contains all semantic information for translation, and the decoder input in the NART model can be considered as a middle layer between input and output. It is not clear how much gain can be achieved by developing a sophisticated submodule for a middle layer in a deep neural network. Second, the encoder-decoder-based NART model is already over-parameterized. We believe that such neural network still has great ability and space to be better optimized if we can provide it with stronger and richer signals, for example, from a much better ART model: Once we have a well-trained ART model, we actually know rich information about the contexts to make the prediction at each time step and the natural word alignments between bilingual sentences. All the information could be invaluable towards the improved training of a NART model.

To well leverage an ART model, we use the *hint-based training* framework (Romero et al., 2014; Chen et al., 2017), in which the information from hidden layers of teacher model (referred as *hints*) are used to guide the training process of a student model. However, *hint-based training* was developed for image classification models and it is challenging to define and use hints for translation. First, the translation model is composed of stacked encoder layers, attention layers, and stacked decoder layers. It is not clear how to define hints in such an encoder-decoder framework. Second, the NART and ART models are of different architectures on the decoding stage. It is not obvious how to leverage hints from the teacher to the training of student with a different architecture. We find that directly applying hints used in the classification tasks fails. In this paper, we first investigate the causes of the bad performance of the NART model, and then define hints targeting to solve the problems. According to our empirical study, we find that the hidden states of the NART model differ from the ART model: the positions where the NART model outputs incoherent tokens will have very high hidden states similarity. Also, the attention distributions of the NART model are more ambiguous than those of ART model. Based on these observations, we design two kinds of hints from the hidden states and attention distributions of the ART model, to help the training of the NART model.

We have conducted experiments on the widely used WMT14 English-to-German/German-to-English (En-De/De-En) task and IWSLT14 German-to-English task. For WMT14 En-De task, our proposed method achieves a BLEU score of 25.20 which significantly outperforms the non-autoregressive baseline models and is even comparable to a strong ART baseline, Google's LSTM-based translation model (24.60 Wu et al. (2016)). For WMT14 De-En task, we also achieve significant performance gains, reaching 29.52 in terms of BLEU.

## 2 RELATED WORKS

### 2.1 AUTOREGRESSIVE TRANSLATION

Given a sentence $x = (x_1, \ldots, x_{T_x})$ from the source language, the straight-forward way for translation is to generate the words in the target language $y = (y_1, \ldots, y_{T_y})$ one by one from left to right. This is also known as the autoregressive factorization in which the joint probability is decomposed into a chain of conditional probabilities, as in the Eqn. (1). Deep neural networks are widely used to model such conditional probabilities based on the encoder-decoder framework. The encoder takes the source tokens $(x_1, \ldots, x_{T_x})$ as input and encodes $x$ into a set of context states $c = (c_1, \ldots, c_{T_x})$. The decoder takes $c$ and subsequence $y_{<t}$ as input and estimates $P(y_t | y_{<t}, c)$ according to some parametric function.

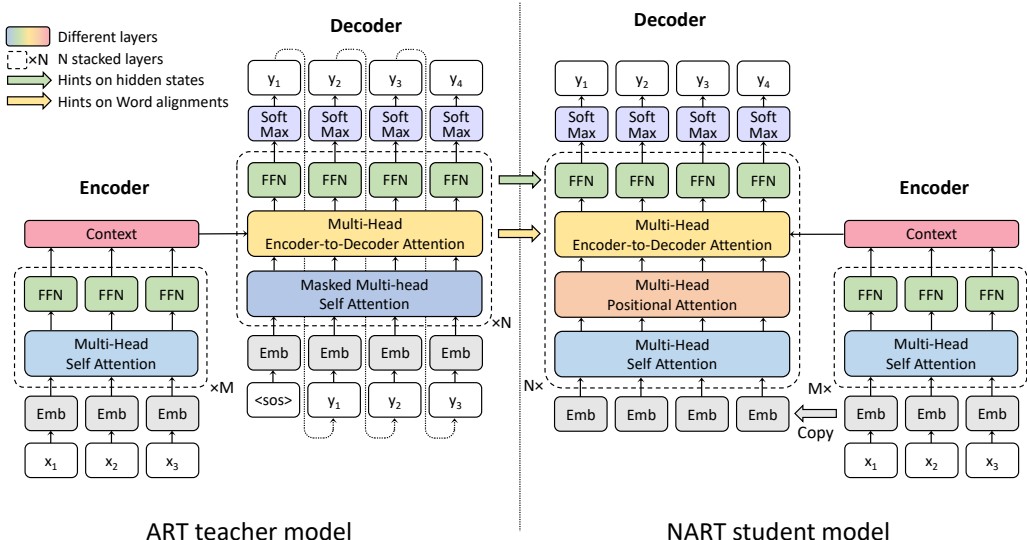

Figure 1: Hint-based training from ART model to NART model.

There are many design choices in the encoder-decoder framework based on different types of layers, e.g., recurrent neural network(RNN)-based (Bahdanau et al., 2014), convolution neural network(CNN)-based (Gehring et al., 2017) and recent self-attention based (Vaswani et al., 2017) approaches. We show a self-attention based network (Transformer) in the left part of Figure 1. While the ART models have achieved great success in terms of translation quality, the time consumption during inference is still far away from satisfactory. During training, the ground truth pair $(x, y)$ is exposed to the model, and thus the prediction at different positions can be estimated in parallel based on CNN or self-attention networks. However, during inference, given a source sentence $x$, the decoder has to generate tokens sequentially, as the decoder inputs $y_{<t}$ must be inferred on the fly. Such autoregressive behavior becomes the bottleneck of the computational time (Wu et al., 2016).

## 2.2 NON-AUTOREGRESSIVE TRANSLATION

In order to speed up the inference process, a line of works begin to develop non-autoregressive translation models. These models follow the encoder-decoder framework and inherit the encoder structure from the autoregressive models. After generating the context states $c$ by the encoder, a separate module will be used to predict the target sentence length $T_y$ and decoder inputs $z = (z_1, \ldots, z_{T_y})$ by a parametric function: $(T_y, z) \sim f_z(x, c; \theta)$, which is either deterministic or stochastic. The decoder will then predict $y$ based on following probabilistic decomposition

$$P(y|x, T_y, z) = \Pi_{t=1}^{T_y} P(y_t|z, c). \tag{3}$$

Different configurations of $T_y$ and $z$ enable the decoder to produce different target sentence $y$ given the same input sentence $x$, which increases the output diversity of the translation models.

Previous works mainly pay attention to different design choices of $f_z$. Gu et al. (2017) introduce *fertilities*, corresponding to the number of target tokens occupied by each of the source tokens, and use a non-uniform copy of encoder inputs as $z$ according to the fertility of each input token. The prediction of fertilities is done by a separated neural network-based module. Lee et al. (2018) define $z$ by a sequence of generated target sentences $y^{(0)}, \ldots, y^{(L)}$, where each $y^{(i)}$ is a refinement of $y^{(i-1)}$. Kaiser et al. (2018) use a sequence of autoregressively generated discrete latent variables as inputs of the decoder.

While the expressiveness of $z$ improved by different kinds of design choices, the computational overhead of $z$ will hurt the inference speed of the NART models. Comparing to the more than $15\times$ speed up in Gu et al. (2017), which uses a relatively simpler design choice of $z$, the speedup of Kaiser et al. (2018) is reduced to about $5\times$, and the speedup of Lee et al. (2018) is reduced to about

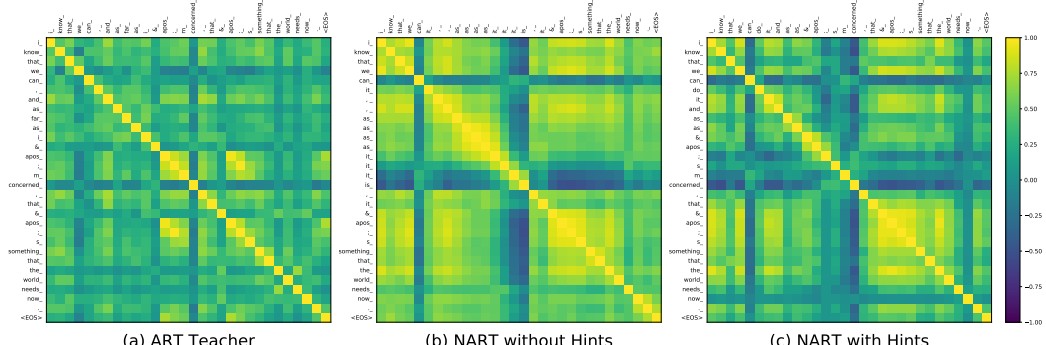

Figure 2: Case study: the above three figures visualize the hidden state cosine similarities of different models. The axes correspond to the generated target tokens. Each pixel shows the cosine similarities $\cos_{ij}$ between the last layer hidden states of the $i$-th and $j$-th generated tokens, where the diagonal pixel will always be 1.0.

$2\times$. This contradicts with the design goal of the NART models: to parallelize and speed up neural machine translation models.

## 3 HINT-BASED TRAINING FOR NON-AUTOREGRESSIVE TRANSLATION

In this section, we introduce the proposed hint-based training algorithm that leverages a well-trained ART model to train the NART model. Our model mostly follows Transformer (Vaswani et al., 2017), with an additional *positional attention* layer proposed by Gu et al. (2017), as shown in the right part of Figure 1. To avoid overhead, we use simple linear combinations of source token embeddings as $z$, which has no learnable parameters. Details about the model can be found in the appendix. We first describe the observations we find about the ART and NART models, and then discuss what kinds of information can be used as hints and how to use them to help the training of the NART model.

### 3.1 OBSERVATION: ILLED STATES AND ATTENTIONS

According to the case study in Gu et al. (2017) and the observations based on our trained model, the translations of the NART models contain incoherent phrases and miss meaningful tokens on the source side. As shown in Table 3, these patterns do not commonly appear in ART models. We aim to answer why the NART model tends to produce incoherent phrases (e.g. repetitive words) and miss relevant translations.

To study the first problem, we visualize the cosine similarities between decoder hidden states of a certain layer in both ART and NART models for sampled cases. Mathematically, for a set of hidden states $r_1, \ldots, r_T$, where $T$ is the number of positions, the pairwise cosine similarity can be derived by $\cos_{ij} = \langle r_i, r_j \rangle / (\|r_i\| \cdot \|r_j\|)$. We then plot the heatmap of the resulting matrix $\cos$, and a typical example is shown in Figure 2.

From the figure, we can see that the cosine similarities between the hidden states at different positions in the NART model are larger than those of the ART model, which indicates that the hidden states across positions in the NART model are "similar". Positions with highly-correlated hidden states are more likely to generate the same word and make the NART model output repetitive tokens, e.g., the yellow area on the top-left of Figure 2(b). However, this problem does not happen in the teacher model. According to our statistics, 70% of the cosine similarities between hidden states in the teacher model are less than 0.25, and 95% are less than 0.5.

To study the second problem, we visualize the encoder-decoder attentions for sampled cases. Good attentions between the source and target sentences are usually considered to lead to accurate translation while poor ones may cause bad translation with wrong tokens. As shown in Figure 3, the attentions of the ART model almost covers all source tokens, while the attentions of the NART model do not cover "farm" but with two "morning". This directly makes the translation result worse

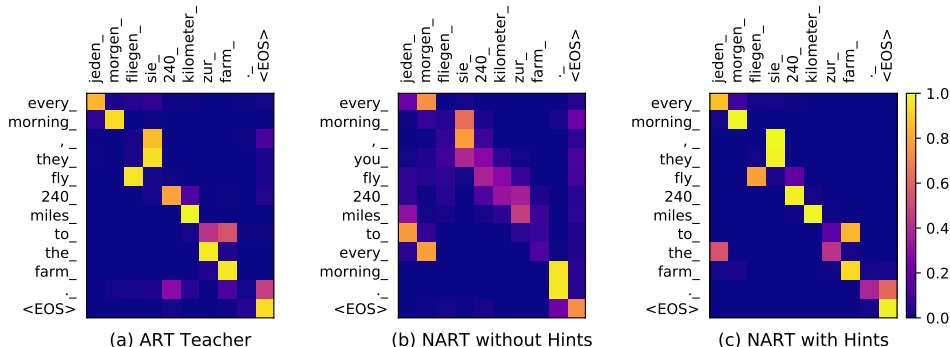

Figure 3: Case study: the above three figures visualize the encoder-decoder attention weights of different models. The x-axis and y-axis correspond to source and generated target tokens respectively. The attention distribution is from a single head of the third layer encoder-decoder attention, which is the most informative one according to our observation. Each pixel shows attention weights $\alpha_{ij}$ between the $i$-th source token and $j$-th target token.

in the NART model. These phenomena inspire us to use the intermediate hidden information in the ART model to guide the learning process of the NART model.

### 3.2 HINT-BASED TRAINING WITH AUTOREGRESSIVE TEACHER

The empirical study in the previous section motivates us to leverage intermediate hidden information from a teacher translation model to help the training of a student model, which is usually referred to as *hint-based training*. Hint-based training (Romero et al., 2014; Wang et al., 2016; Chen et al., 2017) is popularly used to transfer complicated nonlinear mappings from one convolutional neural network to another. In our scenario, we focus on how to define hints from a well-trained ART *teacher* model and use it to guide the training process of a NART *student* model. We study layer-to-layer hints and assume both the teacher model and the student model have an $M$-layer encoder and an $N$-layer decoder, despite that the stacked components are quite different.

Without loss of generality, we discuss our proposed method on a given paired sentence $(x, y)$. In real experiments, losses are averaged over all training data. For the teacher model, we use $a_{t,l,h}^{tr}$ as the encoder-to-decoder attention distribution of $h$-th head in the $l$-th decoder layer at position $t$, and use $r_{t,l}^{tr}$ as the output of the $l$-th decoder layer after feed forward network at position $t$. Correspondingly, $a_{t,l,h}^{st}$ and $r_{t,l}^{st}$ are used for the student model. We propose a hint-based training framework that contains two kinds of hints: hints from hidden states and hints from word alignments.

**Hints from hidden states** The discrepancy of hidden states between ART and NART models motivates us to use hidden states of ART model as a hint for the learning process of the NART model. One of the straight-forward methods is to regularize the $L_1$ or $L_2$ distance between each pair of hidden states in ART and NART models. However, since the decoder input and network components are completely different in ART and NART models, we find using straight-forward regression method on hidden states hurts the learning of the translation model and fails. Therefore, we design a more implicit loss to help the student refrain from the incoherent translation results by acting towards the teacher in the hidden-state level. Specifically, we have

$$\mathcal{L}_{hidden} = \frac{2}{(T_y - 1)T_y N} \sum_{s=1}^{T_y-1} \sum_{t=s+1}^{T_y} \sum_{l=1}^{N} \phi(\cos(r_{s,l}^{st}, r_{t,l}^{st}), \cos(r_{s,l}^{tr}, r_{t,l}^{tr})), \tag{4}$$

where $\phi$ is a penalty function. In particular, we let

$$\phi(d_{st}, d_{tr}) = \begin{cases} -\log(1 - d_{st}), & \text{if } d_{st} \geq \gamma_{st} \text{ and } d_{tr} \leq \gamma_{tr}; \\ 0, & \text{else}, \end{cases} \tag{5}$$

where $-1 \leq \gamma_{st}, \gamma_{tr} \leq 1$ are two thresholds controlling whether to penalize or not. We design this loss since we only want to penalize hidden states that are highly similar in the NART model, but

not similar in the ART model. We have tested several alternative choices of $-\log(1 - d_{st})$, e.g., $\exp(d_{st})$, from which we find similar experimental results.

**Hints from word alignments**    Attention mechanism greatly boosts the performance of the ART models (Bahdanau et al., 2014) and becomes a crucial building block. Many papers discover that the attentions provide reasonable word/phrase alignments between the source and target lead to better performance when predicting target tokens. As we observe that meaningful words in the source sentence are sometimes untranslated by the NART model, and the corresponding positions often suffer from ambiguous attention distributions as shown in Figure 3, we use the word alignment information from the ART model to help the training of the NART model.

In particular, we minimize KL-divergence between the per-head encoder-to-decoder attention distributions of the teacher and the student to encourage the student to have similar word alignments to the teacher model, i.e.

$$\mathcal{L}_{align} = \frac{1}{T_y NH} \sum_{t=1}^{T_y} \sum_{l=1}^{N} \sum_{h=1}^{H} D_{\mathrm{KL}}(a_{t,l,h}^{tr} \| a_{t,l,h}^{st}). \tag{6}$$

Our final training loss $\mathcal{L}$ is a weighted sum of two parts stated above and the negative log-likelihood loss $\mathcal{L}_{nll}$ defined on bilingual sentence pair $(x, y)$, i.e.

$$\mathcal{L} = \mathcal{L}_{nll} + \lambda \mathcal{L}_{hidden} + \mu \mathcal{L}_{align}, \tag{7}$$

where $\lambda$ and $\mu$ are hyperparameters controlling the weight of different loss terms.

## 4    EXPERIMENTS

### 4.1    EXPERIMENTAL SETTINGS

We evaluate our methods on two widely used public machine translation datasets: IWSLT14 German-to-English (De-En) (Huang et al., 2017; Bahdanau et al., 2016) and WMT14 English-to-German (En-De) dataset (Wu et al., 2016; Gehring et al., 2017). IWSLT14 De-En is a relatively smaller dataset comparing to WMT14 En-De. To compare with previous works, we also reverse WMT14 English-to-German dataset and obtain WMT14 German-to-English dataset.

We pretrain Transformer (Vaswani et al., 2017) as the autoregressive teacher model on each dataset. The teacher models achieve 33.26/27.30/31.29 in terms of BLEU in IWSLT14 De-En, WMT14 En-De, De-En test set, respectively. The student model shares the same number of layers in encoder/decoder, size of hidden states/embeddings and number of heads as the teacher models in each task. Following Gu et al. (2017); Kim & Rush (2016), we replace the target sentences in all datasets by the decoded output of the teacher models.

Hyperparameters for hints based training $(\gamma_{st}, \gamma_{tr}, \lambda, \mu)$ are determined to make the scales of three loss components similar after initialization. We also employ label smoothing of value $\epsilon_{ls} = 0.1$ (Szegedy et al., 2016) in all experiments. We use Adam optimizer and follow the optimizer setting and learning rate schedule in Vaswani et al. (2017). Models for WMT14/IWSLT14 tasks are trained on 8/1 NVIDIA M40 GPUs respectively. We implement our model based on the open-sourced `tensor2tensor` (Vaswani et al., 2018) and plan to release it in the near future. More experimental settings can be found in the appendix.

### 4.2    INFERENCE

During training, $T_y$ does not need to be predicted as the target sentence is given. During testing, we have to predict the length of the target sentence for each source sentence. In many languages, the length of the target sentence can be roughly estimated from the length of the source sentence. For example, if the source sentence is very long, its translation is also a long sentence. We provide a simple method to avoid the computational overhead, which uses input length to determine target sentence length: $T_y = T_x + C$, where $C$ is a constant bias determined by the average length differences between the source and target sentences in the training data. We can also predict the target length ranging from $[(T_x + C) - B, (T_x + C) + B]$, where $B$ is the halfwidth. By doing this, we can obtain multiple translation results with different lengths.

Table 1: Performance on the testsets of WMT14 En-De, De-En and IWSLT14 De-En tasks. "/" means the result is not reported. LSTM-based results are from Wu et al. (2016); Bahdanau et al. (2016). ConvS2S results are from Gehring et al. (2017); Edunov et al. (2017). Transformer (Vaswani et al., 2017) results are based on our own reproduction, and are used as the teacher models for NART models. FT: Fertility based NART model by Gu et al. (2017). LT: Latent Transformer by Kaiser et al. (2018). IR: Iterative Refinement based NART model by Lee et al. (2018).

| | WMT14 | | IWSLT14 | | |
| Models | En-De | De-En | De-En | Latency | Speedup |
|---|---|---|---|---|---|
| *Autoregressive models* | | | | | |
| LSTM-based S2S | 24.60 | / | 28.53 | / | / |
| ConvS2S | 26.43 | / | 32.84 | / | / |
| Transformer | 27.30 | 31.29 | 33.26 | 784 ms[‡] | 1.00× |
| *Non-autoregressive models* | | | | | |
| FT | 17.69 | 20.62 | / | 39 ms[†] | 15.6× |
| FT (rescoring 10 candidates) | 18.66 | 22.41 | / | 79 ms[†] | 7.68× |
| FT (rescoring 100 candidates) | 19.17 | 23.20 | / | 257 ms[†] | 2.36× |
| IR (adaptive refinement steps) | 21.54 | 25.43 | / | / | 2.39× |
| LT | 19.8 | / | / | 105 ms[†] | 5.78× |
| LT (rescoring 10 candidates) | 21.0 | / | / | / | / |
| LT (rescoring 100 candidates) | 22.5 | / | / | / | / |
| **NART w/ hints** | **21.11** | **25.24** | **25.55** | **26 ms[‡]** | **30.2×** |
| **NART w/ hints** ($B = 4$, 9 candidates) | **25.20** | **29.52** | **28.80** | **44 ms[‡]** | **17.8×** |

Once we have multiple translation results, we additionally use our ART teacher model to evaluate each result and select the one that achieves the highest probability. As the evaluation is fully parallelizable (since it is identical to the parallel training of the ART model), this rescoring operation will not hurt the non-autoregressive property of the NART model.

We use BLEU score (Papineni et al., 2002) as our evaluation measure. During inference, we set $C$ to $2, -2, 2$ for WMT14 En-De, De-En and IWSLT14 De-En datasets respectively, according to the average lengths of different languages in the training sets. When using the teacher to rescore, we set $B = 4$ and thus have 9 candidates in total. We also evaluate the average per-sentence decoding latencies on one NVIDIA TITAN Xp GPU card by decoding on WMT14 En-De test sets with batch size 1 for our ART teacher model and NART models, and calculate the speedup based on them.[1]

### 4.3 EXPERIMENTAL RESULTS

We compare our model with several baselines: LSTM-based, convolution-based, self attention-based ART models, the fertility based (FT) NART model, the deterministic iterative refinement based (IR) NART model, and the Latent Transformer (LT) which is not fully non-autoregressive by incorporating an autoregressive sub-module in the NART model architecture. The experimental results are shown in the Table 1.

Across different datasets, our method achieves state-of-the-art performances with significant improvements over previous proposed non-autoregressive models. Specifically, our method outperforms fertility based NART model with 6.54/7.11 BLEU score improvements on WMT En-De and De-En tasks in similar settings. Comparing to the ART models, our method achieves comparable results with state-of-the-art LSTM-based sequence-to-sequence model on WMT En-De task. Apart from the translation accuracy, our model achieves a speedup of 30.2 (output a single sentence) or 17.8 (teacher rescoring) times over the ART counterparts. Note that our speedups significantly

---

[1]In Table 1, ‡ and † indicate that the latency is measured on our own platform or by previous works, respectively. Please note that the latencies may be evaluated under different hardware settings and such absolute values are not fair for direct comparison. We also list the rate of speedup of different approaches compared to their ART baselines, which we believe is a better measure of efficiency across different settings.

Table 2: Ablation studies on IWSLT14 De-En. Results are BLEU scores without teacher rescoring.

| Model | $\mathcal{L}_{nll}$ | $\mathcal{L}_{nll} + \mathcal{L}_{align}$ | $\mathcal{L}_{nll} + \mathcal{L}_{align} + \mathcal{L}_{hidden}$ |
|---|---|---|---|
| **BLEU** | 23.08 | 24.76 | **25.55** |

outperform all previous works, because of our lighter design of the NART model: without any computationally expensive module trying to improve the expressiveness.

We provide some case studies for the NART models with and without hints in Table 3. More cases can be found in the appendix. From the first case, we can see that the model without hints translates the meaning of *"as far as I'm concerned"* to a set of meaningless tokens. In the second case, the model without hints omits the phrase *"the farm"* and replaces it with a repetitive phrase *"every morning"*. In the third case, the model without hints mistakenly puts the word *"uploaded"* to the beginning of the sentence, whereas our model correctly translates the source sentence. In all cases, hint-based training helps the NART model to generate better target sentences.

Table 3: Cases on IWSLT14 De-En.

| | |
|---:|:---|
| *Source:* | ich weiß , dass wir es können , und soweit es mich betrifft ist das etwas , was die welt jetzt braucht . |
| *Target:* | i know that we can , and as far as i 'm concerned , that 's something the world needs right now . |
| *ART:* | i know that we can , and as far as i 'm concerned , that 's something that the world needs now . |
| *NART w/o Hints:* | i know that we can it , , as as as as it it it is , it 's something that the world needs now . |
| *NART w/ Hints:* | i know that we can do it and as as 's m concerned , that 's something that the world needs now . |
| *Source:* | jeden morgen fliegen sie 240 kilometer zur farm . |
| *Target:* | every morning , they fly 240 miles into the farm . |
| *ART:* | every morning , they fly 240 miles to the farm . |
| *NART w/o Hints:* | every morning , you fly 240 miles to every morning . |
| *NART w/ Hints:* | every morning , they fly 240 miles to the farm . |
| *Source:* | aber bei youtube werden mehr als 48 stunden video pro minute hochgeladen . |
| *Target:* | but there are over 48 hours of video uploaded to youtube every minute . |
| *ART:* | but on youtube , more than 48 hours of video are uploaded per minute . |
| *NART w/o Hints:* | but on youtube , uploaded than 48 hours hours of video per minute . |
| *NART w/ Hints:* | but on youtube , more than 48 hours video are uploaded per minute . |

We also visualize the hidden state cosine similarities and attention distributions for the NART model with hint-based training, as shown in Figure 2(c) and 3(c). With hints from hidden states, the hidden states similarities of the NART model decrease in general, and especially for the positions where the original NART model outputs incoherent phrases. The attention distribution of the NART model after hint-based training is more similar to the ART teacher model and less ambiguous comparing with the NART model without hints.

Finally, we study the effectiveness of different parts and compare it with a NART model without hints. We conduct an ablation study on IWSLT14 De-En task and the results are shown in Table 2. The hints from word alignments provide an improvement of about 1.6 BLEU points, and the hints from hidden states improve the results by about 0.8 points in terms of BLEU.

## 5 CONCLUSION

Non-autoregressive translation (NART) models have suffered from low-quality translation results. In this paper, we proposed to use hints from well-trained autoregressive translation (ART) models to enhance the training of NART models. Our results on WMT14 En-De and De-En significantly outperform previous NART baselines, and achieve comparable accuracy to an LSTM-based ART model, with one order of magnitude faster in inference. In the future, we will focus on designing new architectures and new training methods for NART models to achieve comparable accuracy as the state-of-the-art ART models such as Transformer.

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

# A  NETWORK ARCHITECTURE

The NART model is developed using the general encoder-decoder framework which is the same as the ART models. Figure 1 shows the architecture of an ART and a NART model based on Transformer. We introduce several components that could help in understanding our proposed hint-based training algorithm. More details about general architectures can be found in Vaswani et al. (2017); Gu et al. (2017).

**Encoder and decoder**  Same as the ART model, the encoder of the NART model takes the embeddings of source tokens as inputs[2] and generates a set of context vectors. As discussed in Section 2.2, the NART model needs to predict $z$ given length $T_y$ and source sentence $x$. We use a simple and efficient method to predict $z = (z_1, \ldots, z_{T_y})$. Given source sentence $x = (x_1, \ldots, x_{T_x})$ and target length $T_y$, we denote $e(x_i)$ as the embedding of $x_i$. We linearly combine the embeddings of all the source tokens to generate $z$ as follows:

$$z_j = \sum_i w_{ij} \cdot e(x_i), \tag{8}$$

where $w_{ij}$ is the normalized weight that controls the contribution of $e(x_i)$ to $z_j$ according to

$$w_{ij} \propto \exp\left(-(j - j'(i))^2/\tau\right), \quad j = 1, \ldots, T_y. \tag{9}$$

where $j'(i) = (T_y/T_x) \cdot i$. $\tau$ is a hyperparameter to control the "sharpness" of the weight distribution. We use $f_z(x, T_y, \tau)$ for this weighted average function to be consistent as in the Eqn. (3).

**Three types of multi-head attention**  The ART and NART models share two types of multi-head attentions: multi-head *self* attention and multi-head *encoder-to-decoder* attention. The NART model specifically uses multi-head *positional* attention to model local word orders within the sentence (Vaswani et al., 2017; Gu et al., 2017). A general attention mechanism can be formulated as querying a dictionary with key-value pairs (Vaswani et al., 2017), e.g.,

$$\text{Attention}(Q, K, V) = \text{softmax}\left(\frac{QK^T}{\sqrt{d_{model}}}\right) \cdot V, \tag{10}$$

where $d_{model}$ is the dimension of hidden representations and $Q$ (Query), $K$ (Key), $V$ (Value) differ among three types of attentions. For self attention, $Q$, $K$ and $V$ are hidden representations of the previous layer. For encoder-to-decoder attention, $Q$ is hidden representations of the previous layer, whereas $K$ and $V$ are context vectors from the encoder. For positional attention, positional embeddings are used as $Q$ and $K$, and hidden representations of the previous layer are used as $V$. The multi-head variant of attention allows the model to jointly attend to information from different representation subspaces, and is defined as

$$\text{Multi-head}(Q, K, V) = \text{Concat}(\text{head}_1, \cdots, \text{head}_H)W^O, \tag{11}$$

$$\text{head}_h = \text{Attention}(QW_h^Q, KW_h^K, VW_h^V), \tag{12}$$

where $W_h^Q \in \mathbb{R}^{d_{model} \times d_k}, W_h^K \in \mathbb{R}^{d_{model} \times d_k}, W_h^V \in \mathbb{R}^{d_{model} \times d_k}$, and $W_h^O \in \mathbb{R}^{d_{model} \times Hd_v}$ are project parameter matrices, $H$ is the number of heads, and $d_k$ and $d_v$ are the numbers of dimensions.

In addition to multi-head attentions, the encoder and decoder also contain fully connected feed-forward network (FFN) layers with ReLU activations, which are applied to each position separately and identically. Compositions of self attention, encoder-to-decoder attention, positional attention, and position-wise feed-forward network are stacked to form the encoder and decoder of the ART model and the NART model, with residual connections (He et al., 2016) and layer normalization (Ba et al., 2016).

---

[2]Following Vaswani et al. (2017); Gu et al. (2017), we also use *positional embedding* to model relative correlation between positions and add it to word embedding in both source and target sides. The positional embedding is represented by a sinusoidal function of different frequencies to encode different positions. Specifically, the positional encoding $e_{pos}$ is computed as $e_{pos}(j, k) = \sin(j/10000^{k/d})$ (for even $k$) or $\cos(j/10000^{k/d})$ (for odd $k$), where $j$ is the position index and $k$ is the dimension index of the embedding vector.

## B  EXTRA RELATED WORKS ON KNOWLEDGE DISTILLATION AND HINT-BASED TRAINING

Knowledge Distillation (KD) was first proposed by Hinton et al. (2015), which trains a small *student network* from a large (possibly ensemble) *teacher network*. The training objective of the student network contains two parts. The first part is the standard classification loss, e.g, the cross entropy loss defined on the student network and the training data. The second part is defined between the output distributions of the student network and the teacher network, e.g, using KL-divergence . Kim & Rush (2016) introduces the KD framework to neural machine translation models. They replace the ground truth target sentence by the generated sentence from a well-trained teacher model. Sentence-level KD is also proved helpful for non-autoregressive translation in multiple previous works (Gu et al., 2017; Lee et al., 2018).

However, knowledge distillation only uses the outputs of the teacher model, but ignores the rich hidden information inside a teacher model. Romero et al. (2014) introduced *hint-based training* to leverage the intermediate representations learned by the teacher model as hints to improve the training process and final performance of the student model. Hu et al. (2018) used the attention weights as hints to train a small student network for reading comprehension.

## C  EXTRA EXPERIMENTAL SETTINGS

**Dataset specifications**  The training/validation/test sets of the IWSLT14 dataset[3] contain about 153K/7K/7K sentence pairs, respectively. The training set of the WMT14 dataset[4] contains 4.5M parallel sentence pairs. Newstest2014 is used as the test set, and Newstest2013 is used as the validation set. In both datasets, tokens are split into a 32000 word-piece dictionary (Wu et al., 2016) which is shared in source and target languages.

**Model specifications**  For the WMT14 dataset, we use the default network architecture of the `base` Transformer model in Vaswani et al. (2017), which consists of a 6-layer encoder and 6-layer decoder. The size of hidden nodes and embeddings are set to 512. For the IWSLT14 dataset, we use a smaller architecture, which consists of a 5-layer encoder, and a 5-layer decoder. The size of hidden states and embeddings are set to 256 and the number of heads is set to 4.

**Hyperparameter specifications**  Hyperparameters $(\tau, \gamma_{st}, \gamma_{tr}, \lambda, \mu)$ are determined to make the scales of three loss components similar after initialization. Specifically, we use $\tau = 0.3, \gamma_{st} = 0.1, \gamma_{tr} = 0.9, \lambda = 5.0, \mu = 1.0$ for IWSLT14 De-En, $\tau = 0.3, \gamma_{st} = 0.5, \gamma_{tr} = 0.9, \lambda = 5.0, \mu = 1.0$ for WMT14 De-En and WMT14 En-De.

**BLEU scores**  We use tokenized case-sensitive BLEU (Papineni et al., 2002)[5] for WMT14 En-De and De-En datasets, and use tokenized case-insensitive BLEU for IWSLT14 De-En dataset, which is a common practice in literature.

## D  EXTRA EXPERIMENTS

**Repetitive words**  According to the empirical analysis, the percentage of repetitive words drops from 8.3% to 6.5% by our proposed hint-based training algorithm on the test set of IWSLT14 De-En, which is a more than 20% reduction. This shows that our proposed method effectively improve the quality of the translation outputs.

**Performance on long sentences**  We select the source sentences whose lengths are at least 40 in the test set of IWSLT14 De-En dataset, and test the models trained with different kinds of hints on this subsampled set. The results are shown in Table 4. As can be seen from the table, our model

---

[3]https://wit3.fbk.eu/

[4]http://www.statmt.org/wmt14/translation-task

[5]BLEU scores are calculated by the scripts at https://github.com/moses-smt/mosesdecoder/blob/master/scripts/generic/multi-bleu.perl

outperforms the baseline model by more than 3 points in term of BLEU (20.63 v.s. 17.48). Note that the incoherent patterns like repetitive words are a common phenomenon among sentences of all lengths, rather than a special problem for long sentences.

Table 4: Performances of different models on long sentences of IWSLT14 De-En test set. Results are BLEU scores without teacher rescoring.

| Model | $\mathcal{L}_{nll}$ | $\mathcal{L}_{nll} + \mathcal{L}_{align}$ | $\mathcal{L}_{nll} + \mathcal{L}_{align} + \mathcal{L}_{hidden}$ |
|---|---|---|---|
| **BLEU** | 17.48 | 19.24 | **20.63** |

## E   EXTRA CASES

Table 5: Extra cases on IWSLT14 De-En.

| | |
|---|---|
| *Source:* | klingt verrückt . aber das geht wieder darauf zurück , dass die eigene vorstellungskraft eine realität schaffen kann . |
| *Target:* | sounds crazy . but this goes back to that theme about your imagination creating a reality . |
| *ART:* | sounds crazy . but this goes back to the point that your imagination can create a reality . |
| *NART w/o Hints:* | sounds crazy . but that back back to that that own imagination can create a reality . |
| *NART w/ Hints:* | sounds crazy . but this goes back to fact that your imagination can create a reality . |
| *Source:* | vor einem jahr oder so , las ich eine studie , die mich wirklich richtig umgehauen hat . |
| *Target:* | i read a study a year or so ago that really blew my mind wide open . |
| *ART:* | one year ago , or so , i read a study that really blew me up properly . |
| *NART w/o Hints:* | a year year or something , i read a study that really really really me me . |
| *NART w/ Hints:* | a year ago or something , i read a study that really blew me me right . |
| *Source:* | wenn ich nun hier schaue , sehe ich die athleten , die in dieser ausgabe erscheinen und die sportarten . |
| *Target:* | so i 'm looking at this ; i see the athletes that have appeared in this issue , the sports . |
| *ART:* | now , when i look here , i see the athletes that appear in this output and the sports species . |
| *NART w/o Hints:* | now if i look at here , i see the athletes that appear in this this ogand the kinds of sports . |
| *NART w/ Hints:* | now if i look at here , i see the athlettes that come out in this output and the species of sports . |
| *Source:* | manchmal eilen diese ideen unserem denken voraus , auf ganz wichtige art und weise . |
| *Target:* | sometimes those ideas get ahead of our thinking in ways that are important . |
| *ART:* | sometimes these ideas forecast our thinking in a very important way . |
| *NART w/o Hints:* | sometimes these ideas make of our thinking , in a very important way . |
| *NART w/ Hints:* | sometimes these ideas make ahead of our thinking in a very important way . |

