# OpenReview forum: "Hint-based Training for Non-Autoregressive Translation"
_ICLR.cc/2019/Conference_

### Official Review · AnonReviewer1 · 2018-11-01
**Some concerns**

**Rating:** 4
**Confidence:** 4

**Review:**

This work proposes a non-autoregressive Neural Machine Translation model which the authors call NART, as opposed to an autoregressive model which is referred to as an ART model. The main idea behind this work is to leverage a well trained ART model to inform the hidden states and the word alignment of NART models. The joint distribution of the targets y given the inputs x, is factorized into two components as in previous works on non-autoregressive MT: an intermediate z which is first predicted from x, which captures the autoregressive part, while the prediction of y given z is non-autoregressive. This is the approach taken e.g., in Gu et al, Kaiser et al, Roy et al., and this also seems to be the approach of this work.  The authors argue that improving the expressiveness of z (as was done in Kaiser et al, Roy et al), is expensive and so the authors propose a simple formulation for z. In particular, z is a sequence of the same length as the targets, where the j^{th} entry z_j is a weighted sum of the embedding of the inputs x (the weights depend in a deterministic fashion on j) . Given this z, the model predicts the targets completely non-autoregressively. However, this by itself is not entirely sufficient, and so the authors also utilize "hints": 1) If the pairwise cosine similarity between two successive hidden states in the student  NART model is above a certain threshold, while the similarity is lower than another threshold in the ART model, then the NART model incurs a cost proportional to this similarity 2) A KL term is used to encourage the distribution of attention weights of the student ART model to match that of the teacher NART model. These two loss terms are used in different proportions (using additional hyperparameters) together with maximizing the likelihood term.

Quality: The paper is not very well written and is often hard to follow in parts. Here are some examples of the writing that feel awkward:

--  Consequently, people start to develop Non-AutoRegressive neural machine
Translation (NART) models to speed up the inference process (Gu et al., 2017; Kaiser et al., 2018;
Lee et al., 2018).

-- In order to speed up to the inference process, a line of works begin to develop non-autoregressive
translation models.

Originality: The idea of using an autoregressive teacher model to improve a non-autoregressive translation model has been used in Gu et al., Roy et al., where knowledge distillation is used. So knowledge distillation paper from Hinton et al., should be cited. Moreover, the authors have missed comparing their work to that of Roy et al. (https://arxiv.org/abs/1805.11063), which greatly improves on the work of Kaiser et al., and almost closes the gap between a non-autoregressive model and an autoregressive model (26.7 BLEU vs 27 BLEU on En-De) while being orders of magnitude faster. So it is not true that:

-- "While the NART models achieve significant speedup during inference (Gu et al., 2017), their accuracy
is considerably lower than their ART counterpart."

-- "Non-autoregressive translation (NART) models have suffered from low-quality translation results"

Significance: The work introduces the idea of using hints for non-autoregressive machine translation. However, I have a technical concern: It seems that the authors complain that previous works like Kaiser et al, Roy et al, use sophisticated submodules to help the expressiveness of z and this was the cause for slowness. However, the way the authors define z seems to have some problems:

- z_j does not depend on z_1, ..., z_{j-1}, so where is the autoregressive dependencies being captured?
- z_1, z_2, ..., z_{T_y} depends only on the length of y, and does not depend on y in any other way. Given x, predicting z is trivial and I don't see why that should help the model f(y | z, x) help at all?
- Given such a trivial z, one can just assume that your model is completely factorial i.e. P(y|x) = \prod_{i} P(y_i|x) since the intermediate z has no information on the y's except it's length.

This is quite suspicious to me, and it seems that if this works, then a completely factorial model should work as well if we only use the "hints" from the ART teacher model. This is a red flag to me, and I am finding this hard to believe.

---

> ### Author Response · Authors · 2018-11-22
> **Author Response**
>
> Thanks for the review! We believe there are some misunderstandings here. We respond to the concerns as below. We will also post our source codes and trained models for verification after the double-blind review period.
>
> 1. Regarding the design choice of z
>
> The reviewer considers that in a non-autoregressive translation model, “z_j does not depend on z_1, ..., z_{j-1}” and “z_1, z_2, ..., z_{T_y} depends only on the length of y” are unreasonable and red flags. However, before our paper, such setting has been shown to work in non-autoregressive translation. We believe the reviewer’s understanding to this might be incorrect.
>
> Gu et al. [1] and we choose to generate z using non-autoregressive ways and ours is a further simplification of [1]. In [1], The hidden z_1, …, z_{T_y} (the “fertility” module) are also mutually independently generated, and have an only limited dependency on y. Please note that the simplicity of z does not mean that the model will definitely suffer from poor translation quality. Although the hidden z is simple, the model itself is a deep neural network, consisting of different components (self attention layer, encoder to decoder attention layer, positional attention layer) that enable the model to learn a complex mapping from x and z to y.
>
> We believe a simple design choice of z is enough and do not list it as a major contribution of our work. Our main technical contribution is to improve the model performance by a more well-designed training algorithm from teachers. Our experimental results show that by our carefully designed training algorithm, a non-autoregressive model with a simple z can achieve near autoregressive performance, while benefiting from the speedup brought by the little overhead of such a simple z.
>
> 2. Regarding "orders of magnitude faster" related works
>
> Thanks for pointing out the recent work from Roy et al. [3], which is also an ICLR submission this year. By checking their paper, we can see that the speedup of Roy et al. [3] is not "orders of magnitude faster". It is only 4.08x when reaching the highest performance, comparing to 17.8x in our work. It is not true that the model by Roy et al. is "orders of magnitude faster".
>
> The main reason is that they choose to use a more complex z using an autoregressive module. Such overhead of z will greatly hurt their speedup, which also contradicts with the initial purpose of introducing non-autoregressive modeling. We believe the translation quality of our model (25.2 for WMT En-De, 29.52 for WMT De-En) is significant given the large speedup of our model.
>
> [1] Gu, Jiatao, et al. "Non-autoregressive neural machine translation." ICLR 2018
> [2] Lee, Jason, Elman Mansimov, and Kyunghyun Cho. "Deterministic Non-Autoregressive Neural Sequence Modeling by Iterative Refinement." EMNLP 2018
> [3] Roy, Aurko, et al. "Theory and Experiments on Vector Quantized Autoencoders." arXiv 2018

---

> ### Author Response · Authors · 2018-11-30
> **Author Response - Regarding Mode Breaking**
>
> Dear reviewer:
>
> To address the concern of mode breaking, we make some discussion here.
>
> 1. To some extent, a machine translation system doesn’t require to model **multiple modes** and is not evaluated by whether the model can generate **multiple modes**.
>
> Machine translation is a real application which aims at providing the correct translation to users, but not providing multiple diverse translations. For example, Google Translate just provides the best translation result from a set of translation candidates, but not a set of translation results with multiple modes. This is a bit different from other generative modeling tasks such as music generation (WaveNet and parallel-WaveNet).
>
> This claim can also be justified from the evaluation metric of machine translation. In the test data, we usually have bilingual sentence pairs, one side is called the source sentence, the other side is called the reference sentence (ground truth target sentence). Sometimes one source sentence has multiple reference sentences. When we have a translation output, the translated sentence will be compared to each reference sentence and use the **maximum** BLEU score as the correctness of the translation. That is being said, although we have multiple references with multiple modes, we just use BLEU score between the translation results and the most similar reference, but do not evaluate the diversity.
>
> 2. Sentence-level knowledge distillation can map  **multiple modes** to a **single mode**.
>
> Our work, Gu et al. and Kaiser et al. (Roy et al.) use sentence-level knowledge distillation(KD), which is super effective in all the works. Sentence-level KD can be considered as using the auto-regressive model output instead of the reference. We have explicitly mentioned this in the paper ("Following Gu et al. (2017); Kim & Rush (2016), we replace the target sentences in all datasets by the decoded output of the teacher models."). As observed by [1], ART model outputs are more stable and the patterns are clearer.
>
> In the example that the reviewer suggests ”suppose our dataset consists of sequences of numbers, where with probability 0.5 the sequence is sorted in ascending order and with probability 0.5 it is sorted in descending order”. We assume the ART model can well capture such modes and further assume there are some training errors: the ascending order sequence is predicted with probability 0.501 while the other one is predicted with probability 0.499. Once we use the greedy search algorithm (e.g., beam search) for distillation, the ART model output is always the in-ascending-order one which has a relatively larger probability.
>
> From the above discussion, we can see that although the original dataset may be **multiple-mode**, the distilled dataset is reduced to **single mode** by using KD. **single mode** dataset is much easier to train non-autoregressive models. That is also a reason why our simple z works.
>
> 3. Regarding appendix
>
> We move the model details to appendix due to the paper length requirements.  Per the reviewer's request, we are willing to move any parts back to the main body if they are considered to be important to the paper quality. Furthermore, we are also willing to release our reproducible codes and models for testing all the tasks mentioned in the paper.
>
> We believe the performance of our non-autoregressive model is significant and we hope the discussions above address the concerns of the reviewers and ACs.
>
> [1] Ott, Myle, Michael Auli, David Granger, and Marc'Aurelio Ranzato. "Analyzing uncertainty in neural machine translation." ICML 2018

---

> > ### Comment · AnonReviewer1 · 2018-12-05
> > **Re: Regarding Mode Breaking**
> >
> > "To some extent, a machine translation system doesn’t require to model **multiple modes** and is not evaluated by whether the model can generate **multiple modes**. "
> >
> > I am fully aware of what Machine Translation is - sure to some extent the MT model is not evaluated on it's ability to capture multiple modes in the data. However, I would argue, that the ability of the model to capture said diversity is an important factor to decide whether to deploy such a model. For example, an imperfect, real world MT model that is unable to capture multiple modes might output an erroneous translation (corresponding to one mode), and the confused user on querying the model again is faced with the same wrong translation because your model is unable to capture the multiple modes in a messy, noisy real-world translation dataset.
> >
> > "Our work, Gu et al. and Kaiser et al. use sentence-level knowledge distillation(KD), which is super effective in all the works. "
> >
> > If you read the papers carefully you will find that sentence-level knowledge distillation was not used in Kaiser et al., but in Roy et al. which you refuse to cite, even though as R3 pointed out is on arxiv since May 2018.
> >
> > Regarding the example I pointed out, as you yourself acknowledge the NART model will be unable to capture the two modes present in the dataset. What if one of the modes that the model misses is the correct one, while the mode it converges on is an incorrect mode due to  noise in the dataset?
> >
> > On the other hand the models of Kaiser et al., Roy et al., are able to capture this diversity because of the autoregressive prior fit on the latents z. While, this model may be able to game the BLEU score which does not evaluate a model on diversity, I still think this approach to non-autoregressive translation is a hack and has limited practical usefulness due to it's inability to sufficiently capture multiple modes in the translation data, due to the inexpressivity of the choice of z's the authors use. Hence my rating still stands.

---

> > > ### Author Response · Authors · 2018-12-08
> > > **Re: Regarding Mode Breaking**
> > >
> > > While we fully get your points, we have to say that we respectfully disagree with your opinions and we are afraid that you have misunderstandings on the research area of neural machine translation.
> > >
> > > - “I would argue, that the ability of the model to capture said diversity is an important factor to decide whether to deploy such a model.”
> > > We have worked on neural machine translation for years and have involved in developing a popular public translation service. We have also communicated with multiple translation teams (including both big Internet companies and small startups).  According to our own experiences and the messages from other groups, the ability to capture such diversity is indeed an unimportant factor to consider while deploying a model, at least at the current stage. According to our knowledge, the most important factors to consider are accuracy, inference latency, and then the model size, which are also hot research topics in neural machine translation. While we agree that the ability to capture diversity is an interesting research problem, it is not the first priority to consider in real-world machine translation systems and not our focus in this work.
> > >
> > > - “an imperfect, real-world MT model that is unable to capture multiple modes might output an erroneous translation (corresponding to one mode)”
> > > While it is not clear whether an erroneous translation is caused by the inability to capture such diversity, we do believe that improving translation accuracy (e.g., in terms of BLEU) can somehow address the problem. This is exactly our focus in this paper.
> > >
> > > - “What if one of the modes that the model misses is the correct one, while the mode it converges on is an incorrect mode due to noise in the dataset?”
> > > Good point. Our model cannot model noise in the dataset. Our model aims to converge to the major mode of the training data (this is also the case for most MT models). If the majority of the translations of a certain source sentence in the training data is incorrect, our model will converge to the incorrect mode and output an incorrect translation. For this case, even if a model can well capture multiple modes, it is still very likely to output an incorrect translation, because the incorrect mode is the major one in the training data and a real-world MT system only output one translation usually corresponding to the major mode. That is, the ability to capture multiple modes *does not* help to solve this problem. Furthermore, if one uses a complete random model for MT, one can eventually get the correct translation result by asking the model repeatedly. However, this kind of “multimode” is definately not we want for a real-world MT model.
> > >
> > > - “this model may be able to game the BLEU score which does not evaluate a model on diversity, I still think this approach to non-autoregressive translation is a hack and has limited practical usefulness due to it's inability to sufficiently capture multiple modes in the translation data”
> > > We are surprised that you say we “game the BLEU” because BLEU “does not evaluate a model on diversity”. According to this judgement rule, most (maybe >90%) research on neural machine translation, including those most influential ones such as LSTM with attention [1], ConvS2S [2] and Transformer [3], will be meaningless, because their primary contribution is to improve the translation quality in terms of BLEU. In other words, they also “game BLEU score which does not evaluate a model on diversity”. We are afraid this statement might be a negative bias towards our work and the whole area of neural machine translation, in which BLEU is the most widely used metric and improving BLEU is one of the most important goals. “BLEU does not reflect diversity” does not mean “BLEU is not a good measure for translation quality”.
> > >
> > >
> > >
> > > [1] Bahdanau D, Cho K, Bengio Y. Neural machine translation by jointly learning to align and translate. ICLR 2015.
> > > [2] Gehring J, Auli M, Grangier D, et al. Convolutional sequence to sequence learning. ICML 2017.
> > > [3] Vaswani A, Shazeer N, Parmar N, et al. Attention is all you need. NeurIPS 2017.

---

### Official Review · AnonReviewer3 · 2018-11-02
**Good results, although knowledge distillation and its use in non-autoregressive NMT should be discussed better.**

**Rating:** 6
**Confidence:** 3

**Review:**

This paper proposes to distill knowledge from intermediary hidden states and
attention weights to improve non-autoregressive neural machine translation.

Strengths:

Results are sufficiently strong. Inference is much faster than for
auto-regressive models, while BLEU scores are reasonably close.

The approach is simple, only necessitating two auxiliary loss functions during
training, and rescoring for inference.

Weaknesses:

The discussion of related work is deficient. Learning from hints is a variant
of knowledge distillation (KD). Another form of KD, using the auto-regressive
model output instead of the reference, was shown to be useful for non-autoregressive
neural machine translation (Gu et al., 2017, already cited). The authors mention using
that technique in section 4.1, but don't discuss how it relates to their work. [1] should
also probably be cited.

Hu et al. [2] apply a slightly different form of attention weight distillation.
However, the preprint of that paper was available just over one month before the
ICLR submission deadline.

Questions and other remarks:

Do the baselines use greedy or beam search?

Why batch size 1 for decoding? With larger batch sizes, the speed-up may be
limited by how many candidates fit in memory for rescoring.

Please fix "are not commonly appeared" on page 4, section 3.1.

[1] Kim, Yoon and Alexander M. Rush. "Sequence-Level Knowledge Distillation" EMNLP. 2016.
[2] Hu, Minghao et al. "Attention-Guided Answer Distillation for Machine Reading Comprehension" EMNLP. 2018

---

> ### Author Response · Authors · 2018-11-22
> **Author Response**
>
> Thanks for the review! We have added discussions on related references and KD to the paper.
>
> In machine translation, the state-of-the-art model uses beam search and thus we follow to use it in the baseline and make comparisons. Batch size 1 for decoding is a common practice when comparing the speedups of non-autoregressive translation models [1, 2, 3].  We follow the practice of previous works to make a fair comparison. Setting batch size 1 and studying the efficiency is also reasonable. Just consider the applications where the translation computation is done on a portable device (e.g. offline translation app on a smartphone). In such a scenario, the user inputs one sentence and expects the translation result.
>
> [1] Gu, Jiatao, et al. "Non-autoregressive neural machine translation." ICLR 2018
> [2] Lee, Jason, Elman Mansimov, and Kyunghyun Cho. "Deterministic Non-Autoregressive Neural Sequence Modeling by Iterative Refinement." EMNLP 2018
> [3] Kaiser, Łukasz, et al. "Fast Decoding in Sequence Models Using Discrete Latent Variables." ICML 2018.

---

### Official Review · AnonReviewer2 · 2018-11-06
**good results, okay paper**

**Rating:** 6
**Confidence:** 4

**Review:**

In this paper, the authors propose an extension to the Non Autoregressive Translation model by Gu et. al, to improve the accuracy of Non autoregressive models as compared to the autoregressive translation models.
The authors propose using hints which can occur as
1. Hidden output matching by incurring a penalty if the cosine distance between the representation differ according to a threshold. The authors state that this reduces same output word repetition which is common for NART models
2. Reducing the KL divergence between the attention distribution of the teacher and the student model in the encoder-decoder attention part of the model.

We see experimental evidence from 3 tasks showing the effectiveness of this technique.

The strengths of this paper are the speedup improvements of using these techniques on the student model while also improving BLEU scores.
The paper is easy to read and the visualisations are useful.

The main issue with this paper is the delta contribution as compared to the NART model is Gu et. al. The 2 techniques, although simple, don't make up for technical novelty.
It would also be good to see more analysis on how much the word repetition reduces using these techniques quantitatively, and performance especially on longer length sequences.

Another issue is the comparison of latency measurements for decoding. The authors state that the hardware and the setting under which the latency measurements are done might be different as compared to previous numbers. Though still impressive speedup improvements, it somehow becomes fuzzy to understand the actual gains.

---

> ### Author Response · Authors · 2018-11-22
> **Author Response**
>
> Thanks for the review! The main contribution of the paper is to show that by our proposed hint-based training algorithm, a simple non-autoregressive model without a complex submodule can reach competitive performance near an autoregressive model, while still being orders of magnitude faster. We think the results and findings are significant and will be helpful to future works in this direction.
>
> We conduct the following two experiments according to the suggestions:
>
> - According to our study, the proposed algorithm reduces the percentage of repetitive words by more than 20% in IWSLT De-En task.
>
> - We filter out all the sentences whose lengths are at least 40 in the test set of IWSLT De-En, and test the baseline model and the model trained with hints on the subsampled set. It turns out that our model outperforms the baseline model by more than 3 points in term of BLEU (20.63 v.s. 17.48). Note that the incoherent patterns like repetitive words are a common phenomenon among sentences of all lengths, rather than a special problem for long sentences.
>
> As stated in the paper, it is quite difficult to find a uniform fair measure for comparing the speed of non-autoregressive models. Since the speedup of non-autoregressive models comes from their fine-grained parallelism, traditional metrics like FLOPs do not fit. Absolute metrics like latency highly depends on underlying hardware and code implementations. Therefore, we use speedup in our paper as a better relative measure for a fair comparison.

---

### Author Response · Authors · 2018-11-22
**Revision to the Paper**

Thanks all reviewers for their valuable comments. We updated a new version of the paper by including the following discussions in the Appendix:

1. We discuss the previous related works on knowledge distillation in Appendix B.

2. We include extra experiments to show the effectiveness of our proposed method in Appendix D.

---

### Meta-Review · Area_Chair1 · 2018-12-17
**needs more work**

**Confidence:** 4
**Recommendation:** Reject

**Metareview:**



+ sufficiently strong results

+ a fast / parallelizable model


- Novelty with respect to previous work is not as great (see AnonReviewer1 and AnonReviewer2's comments)

- The same reviewers raised concerns about the discussion of related work (e.g., positioning with respect to work on knowledge distillation). I agree that the very related work of Roy et al should be mentioned, even though it has not been published it has been on arxiv since May.

- Ablation studies are only on smaller IWSLT datasets, confirming that the hints from an auto-regressive model are beneficial (whereas the main results are on WMT)

-  I agree with R1 that the important modeling details (e.g., describing how the latent structure is generated) should not be described only in the appendix, esp given non-standard modeling choices.  R1 is concerned that a model which does not have any autoregressive components (i.e. not even for the latent state) may have trouble representing multiple modes.  I do find it surprising that the model with non-autoregressive latent state works well however I do not find this a sufficient ground for rejection on its own. However, emphasizing this point and discussing the implication in the paper makes a lot of sense, and should have been done.  As of now, it is downplayed. R1 is concerned that such model may be gaming BLEU: as BLEU is less sensitive to long-distance dependencies, they may get damaged for the model which does not have any autoregressive components.  Again, given the standards in the field, I do not think it is fair to require human evaluation, but I agree that including it would strengthen the paper and the arguments.


Overall, I do believe that the paper is sufficiently interesting and should get published but I also believe that it needs further revisions / further experiments.